# Methyl Jasmonate Induced Oxidative Stress and Accumulation of Secondary Metabolites in Plant Cell and Organ Cultures

**DOI:** 10.3390/ijms21030716

**Published:** 2020-01-22

**Authors:** Thanh-Tam Ho, Hosakatte Niranjana Murthy, So-Young Park

**Affiliations:** 1Institute for Global Health Innovations, Duy Tan University, Danang 550000, Vietnam; hotamqn@gmail.com; 2Department of Botany, Karnatak University, Dharwad 580003, India; hnmurthy60@gmail.com; 3Department of Horticultural Science, Chungbuk National University, Cheongju 28644, Korea

**Keywords:** antioxidant enzyme activity, elicitor, methyl jasmonate, secondary metabolite, signal molecules

## Abstract

Recently, plant secondary metabolites are considered as important sources of pharmaceuticals, food additives, flavours, cosmetics, and other industrial products. The accumulation of secondary metabolites in plant cell and organ cultures often occurs when cultures are subjected to varied kinds of stresses including elicitors or signal molecules. Application of exogenous jasmonic acid (JA) and methyl jasmonate (MJ) is responsible for the induction of reactive oxygen species (ROS) and subsequent defence mechanisms in cultured cells and organs. It is also responsible for the induction of signal transduction, the expression of many defence genes followed by the accumulation of secondary metabolites. In this review, the application of exogenous MJ elicitation strategies on the induction of defence mechanism and secondary metabolite accumulation in cell and organ cultures is introduced and discussed. The information presented here is useful for efficient large-scale production of plant secondary metabolites by the plant cell and organ cultures.

## 1. Introduction

In plant defence systems, each cell has acquired the capability to respond to pathogens and environmental stresses and to build up a defence response. A plant defence mechanism is determined by several factors, mainly depending on their genetic characteristics and physiological state [1,2]. The supplementation of exogenous methyl jasmonate (MJ) to in vitro cultures is responsible for the induction of reactive oxygen species (ROS), regulation of defence response by an accumulation of antioxidant enzyme activity [3,4]. On the other hand, MJ also stimulates molecular signal transduction, regulation of gene expression which leads to accumulation of secondary metabolites [5,6].

Free radicals such as superoxide anion (O_2_^•−^), hydroxyl radical (^•^OH), as well as non-radical molecules like hydrogen peroxide (H_2_O_2_), singlet oxygen (^1^O_2_), are accumulated in plant cells in response to stress mechanism [6,7]. Elicitation with MJ, salicylic acid (SA), environmental stresses, as well as pathogens attack lead to enhanced generation of ROS in plants due to disruption of cellular homeostasis [8,9]. When the level of ROS exceeds the defence mechanisms, a cell is said to be in a state of “oxidative stress” [10]. Although all ROS are extremely harmful to organisms at high concentrations, ROS are well-described second messengers in a variety of cellular processes including tolerance to environmental stresses at low concentration [11]. ROS when present in lower concentration, control many different processes in cells. However, higher concentration ROS in the cell is toxic and detrimental. Therefore, it is necessary for the cells to control the level of ROS tightly to avoid any oxidative injury and not to eliminate them completely [10]. The antioxidants enzymes include superoxide dismutase (SOD), catalase (CAT), guaiacol peroxidase (G-POD), enzymes of ascorbate-glutahione (AsA-GSH) cycle such as ascorbate peroxidase (APX), monodehydroascorbate reductase (MDHAR), dehydroascorbate reductase (DHAR), and glutathione reductase (GR). While ascorbate (AsA), glutathione (GSH), carotenoids, tocopherols, and phenolics serve as potent nonenzymic antioxidants within the cell, were achieved scavenging or detoxification of excess ROS [10].

Application of exogenous MJ to in vitro cultures has emerged as a novel technique for a hyperaccumulation of secondary metabolites. It has been demonstrated that application of MJ to in vitro cultures has induced the antioxidant enzyme activity, expression of defence-related genes, and over-production of secondary metabolites [3,4,12,13,14]. The process or methodology for enhancement of secondary metabolites in *Polygonum multiflorum* adventitious and hairy roots cultures by application of MJ has been presented in Figure 1. The first step is to establish an in vitro culture system, the second step is the optimization of culture conditions for biomass accumulation in the bioreactor culture system. Then, concentration and exposure time of MJ were determined in the third step, thereafter, scale-up of culture process could be achieved up to pilot-scale. HPLC and Fourier-transform infrared spectroscopy (FT-IR) analysis were applied to investigate the quality of bioactive compounds in the culture [15,16,17]. In this review, the role of MJ on the induction of oxidative stress, antioxidant enzyme activity, signal molecules transduction, and secondary metabolite accumulation in plant cell and organ culture system are introduced and discussed.

## 2. Jasmonic Acid (JA) and Methyl Jasmonate (MJ)

The Jasmonic acid (JA) and its derivatives were collectively called Jasmonates (JAs). JAs are cyclopentanone compounds or plant hormones derived from α-linolenic acid. It includes a group of oxygenated fatty acids, and JA is the main precursor of different compounds to this group like methyl jasmonate (MJ), which was first isolated from the essential oil of *Jasminum grandiflorum* [18], and the free acid was isolated subsequently from the culture filtrates of the fungus *Lasiodiplodia theobromae* [19]. JAs was synthesized from fatty acid including three steps. The first step occurs in the membrane chloroplast, where α-linolenic acid and hexadecatrienoic acid were released from membrane phospholipids [20,21]. Lipoxygenase (LOX), allene oxide synthase (AOS), allene oxide cyclase (AOC) are key enzyme of JA biosynthesis in chloroplast and they form (cis-(+)-12-oxophytodienoic acid (OPDA) and dn-OPDA (dinor-OPDA) [22,23]. The second step of JAs synthesis takes place in peroxisome by the process of β-oxidation to give finally jasmonic acid [21,23]. In the final step, JA is exported to the cytoplasm for further modification like MJ, JA-Ile (Figure 2) [21,24,25].

Jasmonic acid (JA) and its precursors and derivatives are known to possess many physiological processes in plant growth and development, and especially the mediation of plant responses to biotic and abiotic stresses [26,27,28]. They have been used extensively for elicitation studies within in vitro culture systems [29]. The elicitation process leads to crosstalk between jasmonates with their receptors which are present in the plasma membrane, which leads an array of defence responses with the cells, including the production of reactive oxygen species (ROS), reactive nitrogen species (RNS), and induction of enzymes of oxidative stress protection [29]. This also leads to synthesis and accumulation of signaling molecules such as JA, salicylic acid (SA), nitric oxide (NO), ethylene (ET) inside the cell and subsequent regulation of gene expression involved in secondary metabolite production [5,30,31]. Therefore, a large number of chemicals such as JA, MJ, SA, ET, are used for elicitation studies in vitro. However, MJ is the most commonly used elicitor which has shown a pronounced effect of accumulation secondary metabolites in plant cell and organ cultures [29,30]. Giri and Zaheer et al. [29] reported that MJ was most abundantly used chemical elicitor (60% of reports) in in vitro culture system, followed by SA and JA (approximately 15 and 10%, respectively). Cell suspension, adventitious roots and hairy roots are commonly used as culture system followed by multiple shoots and embryos for elicitation experiments [4,29,32].

## 3. Usage of Methyl Jasmonate (MJ) on Oxidative Stress and Antioxidant Response

### 3.1. Summary of MJ Elicitation Mechanism

Summary of MJ-treated elicitation mechanism is presented in Figure 3. In the first step, the elicitors are perceived by the specific receptors localized in the plasma membrane, which initiates signalling processes that activate plant defence mechanism [31]. Majority of the studies have revealed that binding of elicitors to receptors leads to induction of pathogenesis-related proteins and the production of ROS and RNS, enzymes of oxidative stress protection, the activation of defence-related genes [30,31,32,33,34,35]. During the process of signal transduction several processes such as protein phosphorylation, lipid oxidation, enhanced antioxidant enzyme activity (SOD, G-POD, APX, CAT), and the activation and the de novo biosynthesis of transcription factors and subsequent expression secondary metabolite biosynthetic genes have been reported by several researchers [5,21].

### 3.2. Oxidative Stress-Induced by Methyl Jasmonate

Recent studies suggest that environmental stresses can increase the oxygen-induced damage to cells due to increased generation of ROS. ROS brings about the peroxidation of membrane lipids, which leads to membrane damage [13,27]. Some studies report that induction of oxidative stress has been observed in plants exposed to MJ and SA [36,37,38]. In agreement with these early studies, exposure to MJ significantly increased the MDA content, an index of lipid peroxidation. O2^−^ is known to be harmful to all membrane constituents (Figure 3 and Figure 4A). The accumulation of MDA content increased significantly in the MJ exposed roots of *Cnidium officinale* compared to the control (Figure 4A). This suggests that MJ lead to the production of O_2_•^−^, •OH resulting in increased lipid peroxidation. In other studies, the better protection in *P. ginseng* seems to result from the more efficient antioxidative system while a significant increase in MDA level in MJ-treated roots appeared to be derived from lower to increased activities of the enzymes [13,14].

### 3.3. Antioxidant Response

MJ is generally considered to modulate many physiological events in higher plants, such as defence responses, flowering and senescence [13,39]. Plants respond to a variety of environmental stresses (biotic and abiotic) through induction of antioxidant defence enzymes that provide protection against further damage [8,40]. Stress tolerance is closely related to the efficiency of antioxidant enzymes and these antioxidant enzymes and metabolites are reported to increase under various environmental stresses [8,12,41]. Numerous studies have demonstrated that MJ plays a key role in the enhancement of antioxidant enzymes activity such as APX, DHAR, MDHAR, AO, GR, GST, GPX, G-POD, CAT, SOD. Ali et al. [12] achieved that induced activity of SOD, G-POD and reduced ascorbate (ASC) contents indicated that antioxidants played an important role for protecting cells from MJ elicitation in *Panax ginseng and Panax quinquefolium* adventitious root cultures. In the apoplast and symplast of roots of sunflower (*Helianthus annuus* L.) seedlings, MJ-elicited roots showed a fast increase in ROS content, followed by a marked increase in the activity of H_2_O_2_-scavenging enzymes, G-POD, APX and CAT. The mechanisms responsible for MJ-induced H_2_O_2_ accumulation was investigated further by studying both the production and scavenging of H_2_O_2_ in the extracellular matrix [42].

The effects of MJ on the accumulation of antioxidant enzyme activity in adventitious root culture of *Cnidium officinale* has been presented in Figure 4B,C. The activity of SOD was significantly decreased in MJ-treated roots, while the accumulation of H_2_O_2_ was increased compared to control (un-treated roots). MJ treatments resulted in a significant increase in G-POD and APX activity which were approximately 1.33 and 1.48-fold higher than the control, respectively. However, the CAT activity was slightly decreased in *C. officinale* adventitious roots treated with MJ. Similar results were also reported by Ali et al. [13] in *P. ginseng* adventitious root cultures.

The main detoxifying enzymes in oxidative stress are SOD, APX, CAT and G-POD, monodehydroascorbate reductase (MDHAR), dehydroascorbate reductase (DHAR), and glutathione reductase (GR) also showed an important role in scavenging stress-induced ROS generated in plants [10,13,14]. These enzymes operate in different subcellular compartments and respond in concert when cells are exposed to oxidative stress [10]. SOD plays a central role in defence against oxidative stress in all aerobic organisms [37]. The enzyme SOD belongs to the group of metalloenzymes and catalyses the dismutation of O_2_•^−^ to O_2_ and H_2_O_2_, which in turn is detoxified by CAT or G-POD or APX reactions (Figure 3). Therefore, the control of the steady-state O_2_•^−^ levels by SOD is an important factor for protecting the cells against oxidative damage. Therefore, SOD is usually considered the first line of defence against oxidative stress [13,14,40,43]. CAT was the first enzyme to be discovered and characterized among antioxidant enzymes. It is a ubiquitous tetrameric heme-containing enzyme that catalyses the dismutation of two molecules of H_2_O_2_ into H_2_O and O_2_. G-POD is a heme-containing protein, preferably oxidizes aromatic electron donor such as guaiacol and pyrogallol at the expense of H_2_O_2_. APX is a central component of AsA-GSH cycle and plays an essential role in the control of intracellular ROS levels. APX uses two molecules of AsA to reduce H_2_O_2_ to water with a concomitant generation of two molecules of MDHA [10,11,44,45,46]. Farooq et al. [47] reported that the application of MJ minimized the oxidative stress, as revealed via a lower level of ROS synthesis in leaves of *Brassica napus*. This study also indicates that MJ plays an effective role in the regulation of multiple transcriptional pathways which were involved in oxidative stress responses, therefore enhanced enzymatic activities and gene expression of important antioxidants (SOD, APX, CAT, POD), secondary metabolites.

### 3.4. Signaling Pathway-Mediated Secondary Metabolite Accumulation in Plant Cell and Organ Cultures

Polyunsaturated fatty acids can produce signals, such as oxylipins, which include JA, JA methyl ester, JA amino acid conjugates, and further JA metabolites [48,49]. JA and its derivatives are known as the signalling molecules that can induce the biosynthetic enzymes involved in the formation of secondary metabolites in ginseng as shown in Figure 5 [5,50]. Zhao et al. [31] reported that exogenous application of MJ could induce endogenous JA biosynthesis, which stimulates expression of saponin biosynthetic genes in ginseng. Kim et al. [51] reported that the expression of farnesyl pyrophosphate synthase (FPS) was induced by MJ and produced the farnesyl diphosphate precursor for squalene biosynthesis. The transcription level of squalene synthase (SS) and squalene epoxidase (SE), key genes in ginseng biosynthesis was enhanced by MJ treatment in *P. notoginseng* and *P. ginseng* [5,51].

The treatment of in vitro cultures with MJ may induce cross-talks between different signal transduction (auxin, ET, ABA, SA, GA, Ca^2+^, O_2_^−^, H_2_O_2_, NO) as in the case in vivo systems. JA signalling pathway triggers *COL1* (*coronatine insensitive1*) gene in *Arabidopsis*, which encode F-box protein and involved in ubiquitination and removal of repressors of the JA signalling pathways [52]. Similarly, two other genes *JAR1* (Jasmonate resistant1) and *JIN1* (Jasmonate insentive1 and *JIN1* are also known as MYC2) were identified in *Arabidopsis*, which are involved in conjugation of jasmonic acid to Ile [53] and transcription regulation of JA-responsive genes, respectively [54]. Another protein *JAZ* (Jasmonate zim domain) mediates the interaction between *JAZ* and *COL1* or other transcriptional factors. COL1 protein, JAZ, and MYC2 constitute the core signal transduction mechanism of JA signalling and also responsible interaction with other transduction pathways [55]. For example, *NtCOL1*, *NtJAZ*, *NtMYC2a/2b* are responsible for nicotine biosynthesis in *Nicotiana tabacum* [55]. JA and auxin singling pathways participate in crosstalk and regulate various plant responses via COL1, MYC2 and JAZ components. When plants are activated with exogenous auxin, the auxin-TIR-AUX/IAA-ARF signalling is activated, which leads to JA synthesis. Contrarily, the endogenous JA induces the expression of auxin synthase gene (*ASA1*) and auxin levels, which leads the regulation of *JAZ1* [55]. During JA and ET crosstalk *ET3* (Ethylene insensitive3, which is involved in ET signalling pathway) and *JAZ-MYC2* (which is involved in the JA signalling pathway) will interact and regulate the stress responses [55]. During JA and ET crosstalk JA and ET may antagonise or coordinately regulated the plant stress responses. At the time of ABA and JA signalling pathways, JAZ-MYC2 participates in the crosstalk between JA and ABA and they regulate the plant responses coordinately. ABA receptor PYL (Pyrabactin resistance1-like proteins) interacts with the JA signalling pathway in many plants [55]. The crosstalk between JA and SA signalling pathways involves many components such as MAPK (mitogen-activated protein kinase), GRX (redox regulators of glutathione) and TRX (thioredoxin). It was demonstrated that JA signalling inhibits SA biosynthesis and accumulation [55]. JA and GA singling pathways regulate plant responses either coordinately or antagonistically. The interaction between JA and SA was brought about by the C-terminus of JAZs with that of MYC2 or DELLA proteins. DELLA interacts with JAZ to release MYC2, leads to activation of MYC2 genes. Concurrently, DELLA interacts with JAZ to inhibit the expression of JA biosynthetic genes via MYB21 and MYB24. Within vitro cultures of ginseng, elicitation with MJ induces crosstalk between MJ and Ca^2+^, O_2_^−^ (superoxide radical), H_2_O_2_, NO, ethylene and JA have been reported [5]. In the beginning, calcium signalling flux is required for further signalling, then both H_2_O_2_ and JA mediate early responses, whereas ethylene production is the late response in elicitor-induced for saponin biosynthesis (Figure 5) [5,56,57].

## 4. Usage of Methyl Jasmonate (MJ) in Plant Cell and Organ Cultures

### 4.1. Application of MJ for Enhancing Secondary Metabolite in In Vitro Culture System

The metabolic profiling of a plant is influenced by various micro-environmental and macro-environmental conditions. The plant responds to its surrounding environmental factors (as biotic and abiotic stress) through the stimulation of secondary metabolism to produce the desired compounds needed for its survival, the process is known as elicitation [58,59,60,61,62]. In recent days, MJ has been used extensively for elicitation studies involving in vitro culture systems such as cell suspension, adventitious root, hairy root and multiple shoot culture system (Table 1). Figure 6 shows the effect of MJ on phenolic compound production in adventitious root culture of *Cnidium officinale*. The adventitious roots treated with 100 µM MJ led to significantly higher yields (two-fold increment) of total phenolic compounds compared with the control treatments. These results corroborate the data for antioxidant enzyme activity (Figure 3 and Figure 4). Elicitor treatment might have resulted in the production of ROS due to stress. In order to mitigate the effects of ROS, plant tissues exhibit a stimulated antioxidant enzyme activity leading to an increased production of secondary metabolites [14,63,64]. Han and Yuan [65] indicated that activation of phenylalanine ammonia-lyase (PAL) activity and accumulation of phenolic compounds is regulated by the oxidative burst in suspension culture of *Taxus cuspidata*, that is believed to change the membrane permeability and lead to the induction of secondary metabolism.

MJ treatment stimulates the biosynthesis of secondary metabolites in plant cell cultures via a large number of control points and triggers the expression of key genes that increase cellular activities at biochemical and molecular levels through the involvement of signal compounds [29]. It also plays a role in the signal transduction, which speeds up enzyme catalysis, thereby leading to the formation of specific compounds such as polyphenol, terpenoids, flavonoids, and alkaloids [4,30]. Ho et al. [62] reported that phenolic levels increased approximately 2-fold in adventitious root samples treated with 50 μM MJ (22.08 mg·g^−1^ DW) versus the control (10.08 mg·g^−1^ DW) in *Polygonum multiflorum*. In addition, hairy root cultures of *P. multiflorum*, the highest total phenolic content (52.87 mg·g^−1^ DW) was 3.4-fold higher than the control, especially, MJ treatment led to significantly higher levels of almost all individual phenolics, such as 1.13-fold increase of physcion, 3.83-fold of quercetin, 1.58-fold of kaempferol, 5.48-fold of p-hydroxybenzoic acid, and 4.3-fold of salicylic acid [62]. Level of ginsenoside accumulation showed a seven-fold enhancement in the adventitious root cultures treated with 100 μM MJ when compared to the control [66]. MJ at 100 µM enhanced the maximum production of xanthotoxin and bergapten at 1.1-fold and 39.6-fold higher than control in *Changium smyrnioides*, respectively [67].

Optimization of MJ concentration, the growth stage and exposure time of cultures are important critical factors for improving secondary metabolite synthesis [29,66]. Elicitation with 100 µM MJ influenced silymarin pigment accumulation in cell culture of *Silybum marimum* after 3 days of treatment [68]. In another report, elicitation with 4 µM MJ for 2 weeks resulted in a 6.5-fold enhancement of solasodine content (9.33 mg g^−1^ DW) than un-elicited hairy root cultures of *Solanum trilobatum* [69]. In addition, increasing of withanolide derivatives accumulation (14-fold) was achieved in 40-day-old hairy roots elicited with 15 µM MJ for 4 h exposure time [70]. In the previous studies on *P. multiflorum*, the phenolic compound in adventitious root culture reached to 2-fold higher than the control after 7-days treatment with 50 µM MJ, whereas 3.4-fold higher was observed in hairy roots after 5-day of exposure time [34,62]. Besides, an increase in 4.5-fold of triterpenoid in hairy root cultures of *Centella asiatica* was reported [71]. Treatment of cultures with 100 µM MJ for 6 days has been reported to influence the accumulation of both phenolic and tanshinones in *Salvia miltiorrhiza* hairy root cultures [72].

### 4.2. Application of MJ for Enhancement of Secondary Metabolites in Bioreactor Cultures

Inhibition of root growth and decreased biomass accumulation are the major challenges with MJ treated cell and organ suspension cultures, this might be due to enhanced accumulation of ROS, which adversely affects root biomass. Airlift bioreactors appear to be ideal for plant cell and organ cultures by efficiently controlling the culture environment, foam generation, shear stress, and oxygen supply [78,83,84]. They are suitable for the cultivation of cell, hairy root, adventitious root and embryo suspension cultures of various medicinal plants [32,78,84,85], and are also applied in MJ treatment to reduce the inhibition of root biomass by two 2-stage cultures. The first step involves the optimization of root growth in bioreactors without elicitor and then concentration and time of elicitor application will have to be standardized in the second step. A numerous study has been reported in using balloon-type bubble bioreactor (BTBB) for both biomass production and enhancement of secondary metabolite treated with MJ. After 6 weeks of culture, the addition of 50 µM MJ for 1 week was found to be the optimal concentration for eleutheroside B and E, and chlorogenic acid production in adventitious root culture of *Eleutherococcus koreanum* [80]. Thanh et al. [80] used 200 µM MJ on day-15 during culture period in cell suspension culture of *Panax ginseng* showed the highest ginsenoside yield after 8 days of treatment. The ginsenoside, Rb group and Rg group content increased 2.9, 3.7, and 1.6-fold after treatment, respectively. In the study of the somatic embryo of *Eleutherococcus senticosus*, the total eleutheroside was increased 7.3-fold after treated with 200 µM MJ [83]. The maximum content of salidroside (4.74 mg·g^−1^ DW) was observed at 125 µM MJ, which was 5-fold higher than the control in *Rhodiola sachalinensis* callus culture [81]. Jiang et al. [80] reported that when added 200 µM MJ to culture medium after 30 days of culture, maximum productivity of bioactive compound was found 8 days after treatment in the adventitious culture of *Olopanax elatus*.

The effect of MJ on biomass and secondary metabolite production in adventitious root culture of *Polygonum multiflorum* and *Echinacea purpurea* is presented in Table 2 and Figure 7. The treatment with MJ inhibited root growth in both medicinal plants after 7 days of treatment. The biomass was decreased by 5.8% (in DW) of *P. multiflorum* and 22.97% (in DW) of *E. purpurea*. The roots also turn to brown with such treatments (Figure 7). However, MJ at 50 µM enhanced the phenolic content approximately 2-fold in *P. multiflorum* compared to non-treated, and also reached to higher the total phenolic compounds when compared to field-grown plants (Table 2). Whereas, total caffeic acid derivatives in *E. purpurea* was also increased 1.16-fold compared to the control with the treatment of 200 µM MJ. Moreover, total productivity was also increased in both plants with 1.92-fold and 1.12-fold compared with non-treated roots.

In addition, the treatment of MJ for improving bioactive compounds is also successfully applied in pilot-scale bioreactors up to 10,000 L [3,84,85,86]. A pilot-scale (500-L) bioreactor cultures were established in adventitious root culture of *Echinacea angustifolia* [87]. In this study, the authors achieved that although root biomass and growth ratio slightly decreased in the 500-L BTBB compared to the 5- and 20-L BTBBs, the highest concentrations of total phenolics (60.41 mg·g^−1^ DW), flavonoids (16.45 mg·g^−1^ DW), and total caffeic acid derivatives (33.44 mg·g^−1^ DW) were observed in the 500-L BTBB when added 100 µM MJ in the culture after 4 weeks. Especially, the accumulation of echinacoside (the major bioactive compound) in MJ-treated adventitious roots grown in the 500-L bioreactor was the highest (12.3 mg·g^−1^ DW), which is approximately three-fold higher than the non-MJ-treated roots cultured in 5- and 20-L bioreactors [80]. Baque et al. [88] achieved the highest accumulation of secondary metabolites in *Morinda citriflora* adventitious root cultures and cultures which were treated with 100 µM MJ after 4 weeks of culture. They reported higher accumulation of anthraquinones (205.75 mg·g^−1^ DW), phenolics (90.26 mg·g^−1^ DW) and flavonoids (93.34 mg·g^−1^ DW) in pilot-scale (500-L) bioreactor cultures when compared to small-scale bioreactor (3-L) cultures (approximately 1.8-fold, 1.3-fold, and 1.55-fold, respectively). In summary, two-stage culture systems should be addressed within in vitro culture system. First, a stage cultivation of cell suspension, adventitious roots and hairy roots for biomass production, followed by addition of MJ (at potential concentration and exposure time) in the second stage for enhancing metabolite production without decreasing root biomass. The application of scale-up bioreactor cultures is promising in the production of biomass and bioactive compounds and such materials/products will provide material for the pharmaceutical and cosmetic industry.

### 4.3. Effect of MJ on Gene Expression and Secondary Metabolite Accumulation

The variation of the expression of genes encoding key enzymes in a biosynthetic pathway directly influences the accumulation of the corresponding secondary metabolites [72,89,90]. The expression levels of the genes responsible for PAL, hydroxycinnamate coenzyme A ligase (4CL), cinnamic acid 4-hydroxylase (C4H), tyrosine aminotransferase (TAT), 4-Hydroxyphenylpyruvate reductase (HPPR), and rosmarinic acid synthase (RAS) were enhanced after 6-days of MJ treatment [72]. These enzymes were involved in the phenolic acid biosynthetic pathways which resulted in 197-fold increment phenolics in *Salvia miltiorrhiza* hairy roots [66]. Rahimi et al. [5] also demonstrated the enhanced regulation of genes involved in MEP (methylerythritol phosphate) and MVA (mevalonate) pathways with the treatment of MJ, in *Panax ginseng* [5]. Genes involved in andrographolide (diterpene) synthesis were influenced by MJ treatment in *Andrographis paniculata* [91]. The MEP pathway genes namely DXS (1-deoxy-d-xylulose 5-phosphate synthase), DXR (1-deoxy-D-xylulose 5-phosphate reductoisomerase), HDS (4-hydroxy-3-methylbut2-enyl-diphosphate synthase) and ISPH (1-Hydroxy-2-methyl-2-(E)-butenyl 4-diphosphate reductase) were up-regulated under MJ stimulation. Similarly, genes involved in the MVA pathway specially HMGS (3-hydroxy-3-methylglutaryl-Co-A synthase) and HMGR (3-hydroxy-3-methylglutaryl-CoA reductase) were also enhanced (Figure 8) by MJ treatment. A number of studies demonstrated that MJ treatment affected on MEP and MVA pathways and secondary metabolites biosynthesis [5,73,92,93].

## Figures and Tables

**Figure 1 ijms-21-00716-f001:**
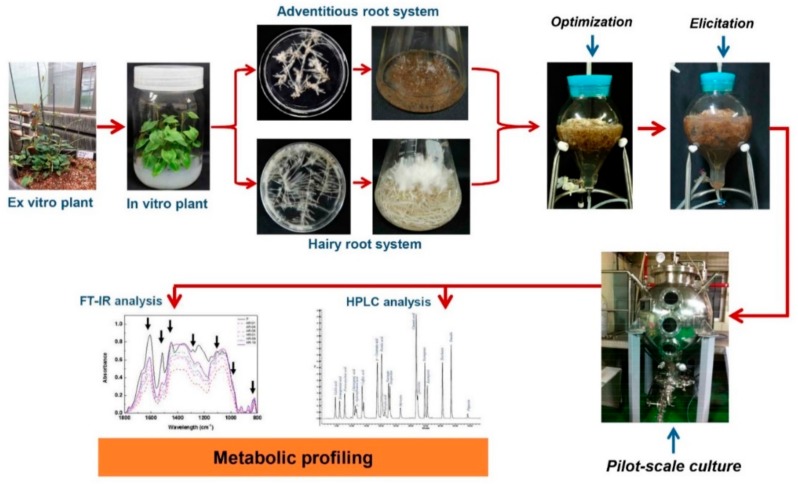
The experimental process of *Polygonum multiflorum* for enhanced production of secondary metabolites.

**Figure 2 ijms-21-00716-f002:**
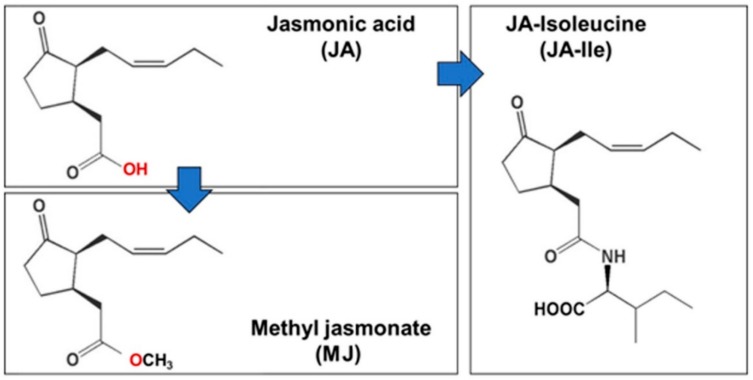
Jasmonic acid and its derivatives.

**Figure 3 ijms-21-00716-f003:**
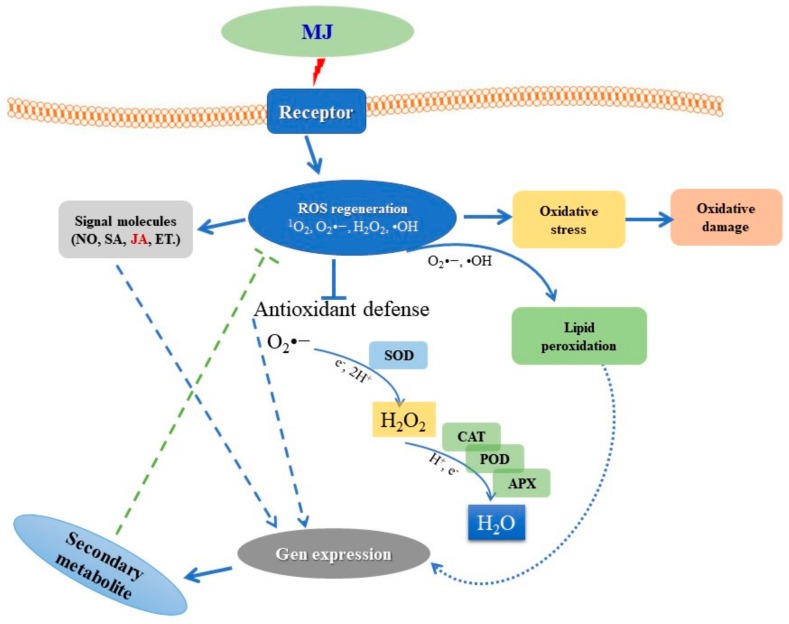
General mechanism after MJ perception. Abbreviations: *ROS* reactive oxygen species, SA salicylic acid, JA jasmonic acid, ET ethylene, O_2_•^−^ superoxide anion, •OH hydroxyl radical, H_2_O_2_ hydrogen peroxide, SOD superoxide dismutase, CAT catalase, G-POD Guaiacol peroxidase, APX ascorbate peroxidase.

**Figure 4 ijms-21-00716-f004:**
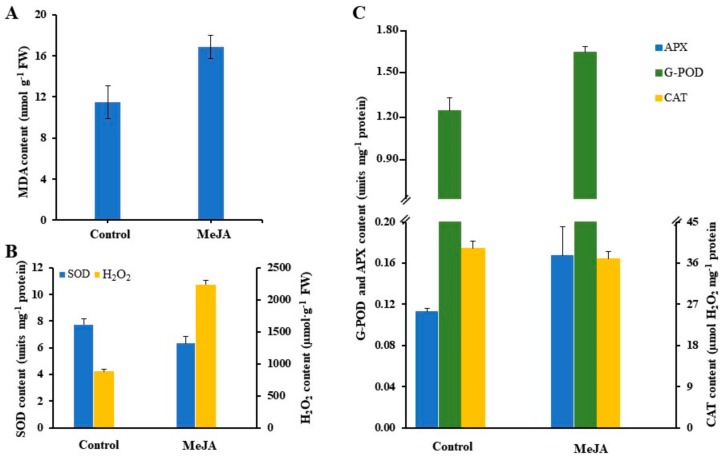
Effect of MJ on malondialdehyde (MDA) content (**A**), and antioxidant enzyme activity (**B**,**C**) in adventitious root cultures of *Cnidium officinale*. *SOD* superoxide dismutase, *CAT* catalase, *G-POD* guaiacol peroxidase, *APX* ascorbate peroxidase. Values represent mean ± SE (*n* = 3).

**Figure 5 ijms-21-00716-f005:**
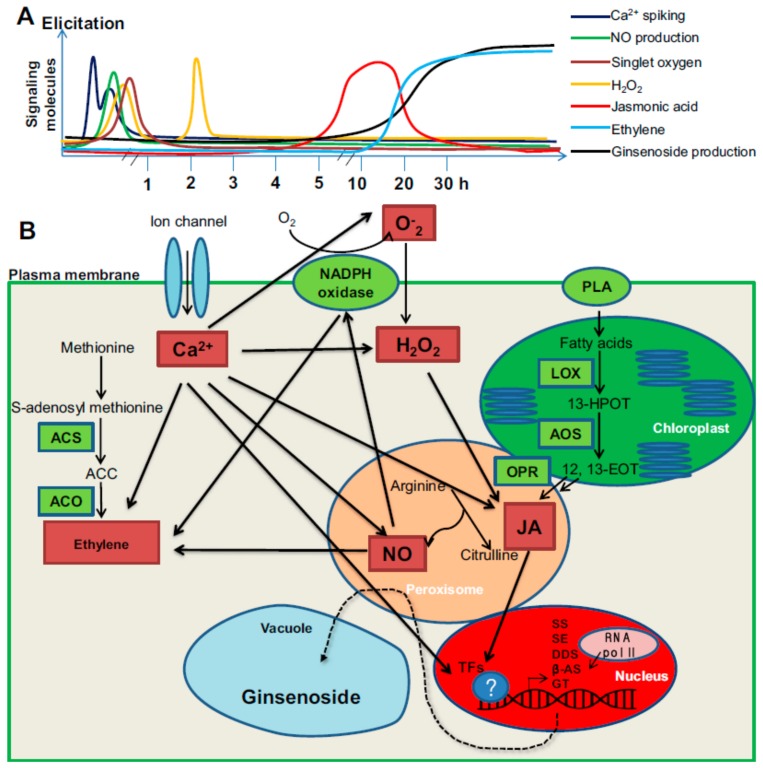
Schematic illustration of the sequential signalling pathways activated in elicited ginseng (**A**). Model of cross-talk between different signal transductions (**B**). Cross talk between different signalling molecules is shown by bold arrows. *ACC* 1-aminocyclopropane-1-carboxylic acid, *ACS* 1-aminocyclopropane-1-carboxylic acid synthase, *ACO* 1-aminocyclopropane-1-carboxylic acid, *AOS* allene oxide synthase, *β-AS* beta-amyrin synthase, *DDS* dammarenediol synthase, *12,13-EOT* 12,13(S)-epoxyoctadecatrienoic acid, *H_2_O_2_* hydrogen peroxide, *13-HPOT* (13S)-hydroperoxyoctadecatrienoic acid, *JA* jasmonic acid, *LOX* lipoxygenase, *NO* nitric oxide, *NOS* nitric oxide synthase, *O_2_^−^* superoxide radical, *OPR* oxophytodienoate reductase, *PLA* phospholipase, *SS* squalene synthase, *SE* squalene epoxidase, *TFs* transcription factors, *UGRdGT* UDPG/ginsenoside Rd glucosyltransferase (Adapted from Rahimi et al. [5]).

**Figure 6 ijms-21-00716-f006:**
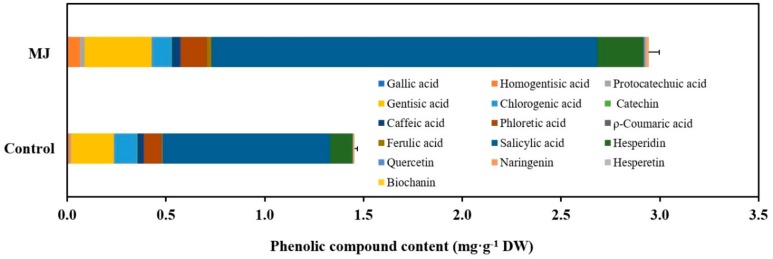
Effect of MJ on phenolic compounds production in *Cnidium officinale* adventitious root culture.

**Figure 7 ijms-21-00716-f007:**
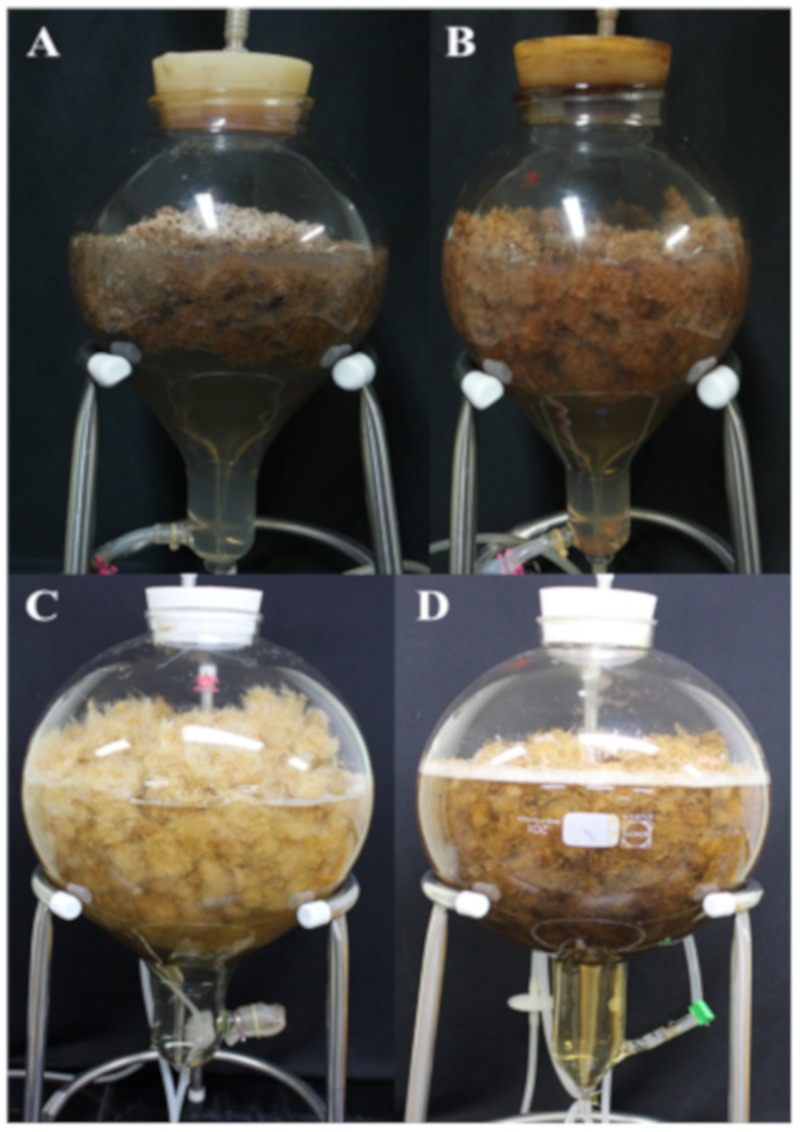
Effect of MJ on biomass production in *Polygonum multiflorum* and *Echinacea purpurea* adventitious root culture system. (**A**,**B**) control and 100 µM MJ in *P. multiflorum*, (**C**,**D**) control and 100 µM MJ in *E. purpurea*, respectively.

**Figure 8 ijms-21-00716-f008:**
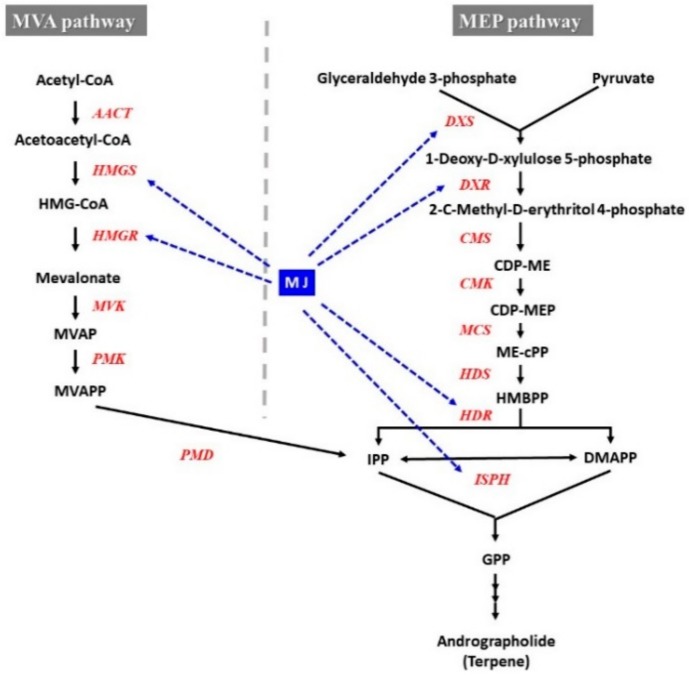
Andrographolide biosynthetic gene activation by MJ elicitation in *Andrographis paniculata* [91]. Dashed blue arrows show the relationship between MJ signalling and andrographolide biosynthetic genes. HMG-CoA, 3-Hydroxy-3-methylglutaryl CoA. MVA, mevalonic acid. MVAP, mevalonic acid 5-phosphate. MVAPP, mevalonic acid 5-diphosphate. IPP, isopentenyl diphosphate. DMAPP, dimethylallyl diphosphate. MEP, 2-C-methyl-D-erythritol 4-phosphate. CDP-ME, 4-diphosphocytidyl-2-C-methyl-d-erythritol. CDP-MEP, 4-diphosphocytidyl2-C-methyl-D-erythritol 2-phosphate. ME-cPP, 2-C-methyl-D-erythritol 2,4-cyclodiphosphate. HMBPP, 1-hydroxy-2-methyl-2-(E)-butenyl 4-diphosphate. Enzymes of the MVA pathway: HMGS, HMG-CoA synthase. HMGR, HMG-CoA reductase. MVK, MVA kinase. PMK, MVAP kinase. PMD, MVAPP decarboxylase. Enzymes of the MEP pathway: DXS, DOXP synthase. DXR, DOXP reductoisomerase. CMS, CDP-ME synthase. CMK, CDP-ME kinase. MCS, ME-2,4cPP synthase. HDS, HMBPP synthase. HDR, HMBPP reductase, ISPH, 1-Hydroxy-2-methyl-2-(E)-butenyl 4-diphosphate reductase.

**Table 1 ijms-21-00716-t001:** Usage of MJ for enhance secondary metabolite within in vitro culture system.

Plant	Material ^z^	Culture System	MJ	Elicitor Duration	Biomass	Target Compound	Fold Increase Compound	Reference
*Panax notoginseng*	CS	Flask	200 µM	7 day	Not significant	Rb, Rg ginsenosides	3.38	[73]
*Silybum marimum*	CS	Flask	100 µM	3 day	Decreased	Silymarin	4	[68]
*Morinda citrifolia*	CS	Flask	150 µM	2 day	Slightly decreased	Anthraquinone	4	[74]
*Mentha* x *piperita*	CS	Flask	100 µM	2 day	Decreased	Rosmarinic acid	1.8	[75]
*Changium smyrnioides*	CS	Flask	100 µM	5 day	Decreased	Xanthotoxin, bergapten	1.1 and 39.6	[67]
*Scopolia parviflora*	AR	Flask	1 mM	1 day	Slightly decreased	Scopolamine and hyoscyamine	2–3	[76]
*Perovskia abrotanoides*	AR	Flask	10 µM	7 day	Not significant	Tanshinone	2.3	[77]
*Polygonum multiflorum*	AR	Flask	50 µM	7 day	Decreased	Phenolic	2	[62]
*Withania somnifera*	HR	Flask	15 µM	4 h	Slightly decreased	Withanolide derivatives	14	[71]
*Solanum trilobatum*	HR	Flask	4 µM	14 day	Slightly decreased	Solasodine	6.5	[70]
*Salvia miltiorrhiza*	HR	Flask	100 µM	6 day	Decreased	Phenolic and tanshinones	3.3 and 2.5	[71]
*Polygonum multiflorum*	HR	Flask	50 µM	5 day	Slightly decreased	Phenolic	3.2	[34]
*Echinacea purpurea*	HR	Flask/Bioreactor	400 µM	7 day	Not significant	Triterpenoid	4.5	[78]
*Polygonum multiflorum*	AR	Bioreactor	50 µM	7 day	Slightly decreased	Phenolic	2.1	[62]
*Panax ginseng*	AR	Bioreactor	100 µM	10 day	Decreased	Ginsenoside	11-fold in Rb group	[66]
*Panax ginseng and Panax quinquefolium*	AR	Bioreactor	200 µM	9 day	Not significant	Saponin	4	[12,13]
*Eleutherococcus koreanum*	AR	Bioreactor	50 µM	7 day	Not significant	Eleutherosides B and E, and chlorogenic acid	1.1, 1.4 and 1.2	[79]
*Olopanax elatus*	AR	Bioreactor	200 µM	8 day	Decreased	Polysaccharide, phenolic	1.83 and 1.49	[80]
*Panax ginseng*	CS	Bioreactor	200 µM	8 day	Decreased	Ginsenoside	1.6–3.7	[81]
*Rhodiola sachalinensis*	CS	Bioreactor	125 µM	7 day	Decreased	Salidroside	5	[82]
*Eleutherococcus senticosus*	SE	Bioreactor	200 µM	7 day	Decreased	Eleutheroside	7.3	[83]

^z^*CS* cell suspension culture, *AR* adventitious root culture, *HR* hairy root culture, *SE* somatic embryos culture.

**Table 2 ijms-21-00716-t002:** Effect of MJ on the accumulation of biomass and secondary metabolites in *Polygonum multiflorum* and *Echinacea purpurea* adventitious root culture system.

Systems	FW (g·L^−1^)	DW (g·L^−1^)	Dry Matter (%)	Growth Ratio ^z^	Total Bioactive Compounds (mg·g^−1^ DW) ^y^	Total Productivity (mg·L^−1^) ^x^
*Polygonum multiflorum* ^w^						
Control	93.95 ± 3.23	10.61 ± 0.32	11.30 ± 0.11	20.23 ± 0.64	11.20 ± 0.17	118.82 ± 1.78
MJ50	90.09 ± 1.51	9.99 ± 0.23	11.10 ± 0.44	18.98 ± 0.46	22.83 ± 0.30	228.08 ± 3.00
5-year-old root					24.78 ± 0.82	
*Echinacea purpurea* ^v^						
Control	83.81 ± 2.12	8.10 ± 0.72	9.63 ± 0.64	15.19 ± 1.45	25.22 ± 1.27	204.16 ± 10.24
MJ200	67.26 ± 1.72	6.24 ± 0.40	9.27 ± 0.53	11.48 ± 0.80	38.32 ± 0.34	239.20 ± 4.44

Data present mean ± SE. ^v^ Adventitious root of *E. purpurea* culture in 20-L bioreactor in 3 weeks, treated with 200 µM MJ for 7 days before harvest. ^w^ Adventitious root of *P. multiflorum* culture in 20-L bioreactor in 3 weeks, treated with 100 µM MJ for 7 days before harvest. ^x^ Total productivity (mg·L^−1^ medium) = Total bioactive compound content (mg·g^−1^ DW) x DW (mg·L^−1^). ^y^ Total bioactive compounds (mg·g^−1^ DW) = Total phenolic compounds in *P. multiflorum*, and total caffeic acid derivatives in *E. purpurea*. ^z^ Growth ratio = (DW—Initial DW)/ Initial DW.

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
