# Peer review of "Methyl Jasmonate Induced Oxidative Stress and Accumulation of Secondary Metabolites in Plant Cell and Organ Cultures"

_ijms, 2020, doi:10.3390/ijms21030716_

Round 1

Reviewer 1 Report

General comments

The English should be checked throughout the text, especially the verb “achieve” should be used properly. Replace MJ with MeJA trought the text.

Specific comments

Line 34: remove “of”;

Line 39: use “with” instead of “such as“;

Lines 44-45: the subject of the following sentence is missing “Because ROS can act as damaging or signalling molecule depends on the delicate equilibrium between ROS production and scavenging”; please rephrase the sentence;

Line 47: replace “including” with “include”;

Line 74: replace “to” with “of”;

Line 81: replace “take” with “takes”;

Line 85: replace “it” with “its” in the capture of legend of Fig. 2;

Lines 98-99: please rephrase for better understanding;

Line 133: “MeJA generally considered to ….”, insert “is” between MeJA and generally;

Line 130,138, 150 and 151: replace MJ with MeJA;

Line 167: replace “pyragallol” with “pyrogallol”;

Line 176: insert “is” between “scavenging and” and “involved”;

Lines 175-177: the following sentence should be rephased “These findings reveal that MeJA improves ROS scavenging and involved in the oxidative stress processes by regulating and through enhanced antioxidant defence system and secondary metabolite”;

Line 190: replace “has been achieved” with “reported”;

Lines 198-199: replace “finally enhanced the secondary metabolites production with “finally, the secondary metabolites production increased”;

Line 217: insert “has been” between “In the recent days, MeJA” and “used”;

Lines 224-226: edit the sentence as follows “In order to mitigate the effects of ROS, plant tissues exhibit a stimulated antioxidant enzyme activity leading to an increased production of secondary metabolites”;

Line 241: replace “which a “with “such as”;

Line 251: replace “elicited” with “elicitation”;

Line 253: edit the sentence as follows “In addition, increasing of withanolide derivatives accumulation (14-fold) was achieved in 40-day-old hairy roots elicited with 15 μM MeJA for 4 h exposure time;

Lines 269-271: rephrase the following sentence for better understanding “Inhibition of root growth and decreased biomass accumulation are the major challenges elicitor treated cell and organ suspension cultures, this might be due to enhanced accumulation of ROS, which adversely affects root biomass”;

Line 275: replace “inhibited” with “inhibition”;

Line 279 and Line 292: replace “subjected” with “treated”;

Lines 350-354: the following sentence should be rephased “This result also showed the expression levels of upstream genes of both the MEP and MVA pathways, as well as that of a downstream gene geranylgeranyl diphosphate synthase (GGPPS), were altered by the addition of MeJA, which affect to tanshinone biosynthesis. MeJA also the regulated in MEP (methylerythritol phosphate) and MVA (mevalonate) pathways, as control ginsenoside accumulation in Panax ginseng”.

Author Response

Response to reviewer’s comments

Reference No: Manuscript ID: ijms-683327
Type of manuscript: Review
Title: Methyl Jasmonate Induced Oxidative Stress and Accumulation of
Secondary Metabolites in Plant Cell and Organ Cultures
Authors: Thanh Tam Ho, Hosakatte Niranjana Murthy, So-Young Park

With reference to the above, authors are thankful to the anonymous reviewers for their valuable comments on manuscript.  We have revised the manuscript in the light of reviewer’s comments and incorporated all the corrections in the revised manuscript. All grammatical and typographical errors have been corrected and incorporated in the revised text.  The corrections incorporated in revised text are highlighted in blue colour. Following are specific corrections incorporated in the revised manuscript.

Reviewer #1

The English should be checked throughout the text, …..

Answer:

i) All the grammatical and typographical errors have been corrected and incorporated in the revised text. The revised manuscript has been reviewed by native English speaker.  

ii) The word ‘MJ’ has been used in place of ‘MeJA’ throughout the text.

iii) All the suggested corrections are incorporated in the revised manuscript.

Reviewer 2 Report

The content of the review sounds interesting and of interest for a large audience. The main limitation is the English that  should be in-depth revised. In fact, the language used is not fluent and often makes it difficult to understand the whole story. Therefore, the manuscript would need to an extensive editing by a native English speaker. Moreover, a final paragraph with Future Prospects about the exploitation of JA for metabolite accumulation in plant cell and organ cultures is required to conclude the review. Specific comments are highlighted in pdf of the manuscript (attached file). 

Author Response

Response to reviewer’s comments

Reference No: Manuscript ID: ijms-683327
Type of manuscript: Review
Title: Methyl Jasmonate Induced Oxidative Stress and Accumulation of
Secondary Metabolites in Plant Cell and Organ Cultures
Authors: Thanh Tam Ho, Hosakatte Niranjana Murthy, So-Young Park

With reference to the above, authors are thankful to the anonymous reviewers for their valuable comments on manuscript.  We have revised the manuscript in the light of reviewer’s comments and incorporated all the corrections in the revised manuscript. All grammatical and typographical errors have been corrected and incorporated in the revised text.  The corrections incorporated in revised text are highlighted in blue colour. Following are specific corrections incorporated in the revised manuscript.

Reviewer # 2

The content of the review …..

Answer:

i) All the grammatical and typographical errors have been corrected and incorporated in the revised text. The revised manuscript has been reviewed by native English speaker.

ii) All the suggested corrections are incorporated in the revised manuscript

Reviewer 3 Report

In Figure 3, Gen expression should be changed to Gene expression.

In Figure 7, picture A and B are much darker than picture C and D. Please adjust brightness.

                  I know AB and CD are not related each other but it will be better for readers.

This review paper focused on MeJA application in cell culture system. It is well written review paper overall.

Please add one more section to compare the advantage of MeJA application in cell culture system compared to other systems (open filed conditions). It is reported that MeJA application increased ethylene production in broccoli and other vegetables. Hence, the MeJA treatment may reduce postharvest quality. In this point of view, MeJA application in cell culture system has advantage compared to using plant to produce secondary metabolites. Readers want to see and learn what is the advantage and disadvantage of cell and organ culture system to produce secondary metabolites.

Author Response

Response to reviewer’s comments

Reference No: Manuscript ID: ijms-683327
Type of manuscript: Review
Title: Methyl Jasmonate Induced Oxidative Stress and Accumulation of
Secondary Metabolites in Plant Cell and Organ Cultures
Authors: Thanh Tam Ho, Hosakatte Niranjana Murthy, So-Young Park

With reference to the above, authors are thankful to the anonymous reviewers for their valuable comments on manuscript.  We have revised the manuscript in the light of reviewer’s comments and incorporated all the corrections in the revised manuscript. All grammatical and typographical errors have been corrected and incorporated in the revised text.  The corrections incorporated in revised text are highlighted in blue colour. Following are specific corrections incorporated in the revised manuscript.

Reviewer # 3

In Figure 3…

Answer: i) Figure 3 has been corrected – ‘Gen expression’ has been changed to ‘Gene expression’.

ii) Figure 7. The pictures A and B are of Echinacea purpurea adventitious root cultures, the morphology of these roots is usually brown which turns black with age of the cultures. Whereas, the pictures of C and D are of Polygonum multiflorum adventitious root cultures – usually adventitious roots of multiflorum are pale-yellow in colour. Therefore, variability with pictures exist. We have incorporated revised figure 7 (pictures) now.

iii) Authors agree with reviewer opinion that MJ application with in vivo (open field cultivated) plants leads ethylene production in some vegetables. However, such a phenomenon was not recorded or reported with in vitro cultures. We did not get sufficient literature in this regard, therefore, these aspects are not included in the conclusion section.

Round 2

Reviewer 2 Report

The manuscript has been improved, however, there are important comments that were not addressed in the current version of the manuscript version: 

-Recent literature about the effect of MeJA in protecting plants against  stresses (oxidative and not only) must be cited. For instance  “Bertini L, Palazzi L, Proietti S, Pollastri S, Arrigoni G, de Laureto P, Caruso C (2019). Proteomic Analysis of MeJa-Induced Defense Responses in Rice against Wounding. International Journal of Molecular Sciences, 20, 2525”. Additionally has been proved that MeJA has a priming effect on plants for augmented expression of a subset of defense-related genes, as reported in "Bertini L, Proietti S, Focaracci F, Sabatini B, Caruso C (2018). Epigenetic control of defense genes following MeJA-induced priming in rice (O. sativa). Journal of Plant Physiology, 228: 166-177".

-In paragraph 3.4 the authors should  discuss in a more exhaustive way ''crosstalk between different signal trasduction..''. Cross-communication is a crucial step for the final outcome of the effect of hormones. 

- In paragraph 4.3. I suggest to I suggest to make a figure where gene expression is correlated with metabolite production of a certain pathway. 

Author Response

Reference No: Manuscript ID: ijms-683327
Type of manuscript: Review
Title: Methyl Jasmonate Induced Oxidative Stress and Accumulation of
Secondary Metabolites in Plant Cell and Organ Cultures
Authors: Thanh Tam Ho, Hosakatte Niranjana Murthy, So-Young Park

With reference to the above, authors are thankful to the anonymous reviewers for their valuable comments on manuscript.  We have revised the manuscript in the light of reviewer’s comments and incorporated all the corrections in the revised manuscript. The corrections incorporated in revised text are highlighted in blue colour. Following are specific corrections incorporated in the revised manuscript.

Query #1. Recent literature about the effect of MeJA ……

Answer: Both references suggested by reviewer are incorporated in the revised manuscript suitablly.

Answer: Both the referneces - Bertini, L.; Palazzi, L.; Proietti, S.; Pollastri, S.; Arrigoni, G.; de Laureto, P.; Carusco, C. Proteomic analysis of MeJa-induced defense responses in rice against wounding. Int. J. Mol. Sci. 2019, 20: 2525. Bertini, L.; Proietti, S.; Focaracci, F.; Sabatin, B.; Carusco, C. Epigenetic control of dense genes following MeJA-induced priming in rice (O. sativa). J. Plant Physiol. 2018, 228: 166-177.

Query #2. In paragraph 3.4 the authors should discuss …. Crosstalk between different signal….

Answer: Crosstalk between different signal transduciton/s have been discussed as per the suggestion.

Query #3. In paragraph 4.3 I suggest to make a figure…

Answer: Figure on gene expression induced by MJ on terpenoid biosynthesis is incorporated as Figure 8.
